# Apple Glycosyltransferase MdUGT73AR4 Glycosylates ABA to Regulate Stomatal Movement Involved in Drought Stress

**DOI:** 10.3390/ijms25115672

**Published:** 2024-05-23

**Authors:** Lijun Mu, Xuekun Wang, Yingxin Ma, Aijuan Zhao, Shibo Han, Ru Li, Kang Lei, Lusha Ji, Pan Li

**Affiliations:** State Key Laboratory for Macromolecule Drugs and Large-Scale Manufacturing, School of Pharmaceutical Sciences, Liaocheng University, Liaocheng 252059, China; mlj15215354741@163.com (L.M.); wangxuekun@lcu.edu.cn (X.W.); m13021553065@163.com (Y.M.); zaj15105403245@163.com (A.Z.); hshibo1999@163.com (S.H.); liru0024@163.com (R.L.); leikang@lcu.edu.cn (K.L.)

**Keywords:** glycosyltransferase, MdUGT73AR4, ABA, stomatal, AREB1B, drought tolerance

## Abstract

Abscisic acid (ABA) is a drought-stress-responsive hormone that plays an important role in the stomatal activity of plant leaves. Currently, ABA glycosides have been identified in apples, but their glycosyltransferases for glycosylation modification of ABA are still unidentified. In this study, the mRNA expression of glycosyltransferase gene *MdUGT73AR4* was significantly up-regulated in mature apple leaves which were treated in drought stress by Real-Time PCR. It was hypothesised that *MdUGT73AR4* might play an important role in drought stress. In order to further characterise the glycosylation modification substrate of glycosyltransferase MdUGT73AR4, we demonstrated through in vitro and in vivo functional validation that MdUGT73AR4 can glycosylate ABA. Moreover, the overexpression lines of *MdUGT73AR4* significantly enhance its drought stress resistance function. We also found that the adversity stress transcription factor AREB1B might be an upstream transcription factor of *MdUGT73AR4* by bioinformatics, EMSA, and ChIP experiments. In conclusion, this study found that the adversity stress transcription factor AREB1B was significantly up-regulated at the onset of drought stress, which in turn positively regulated the downstream glycosyltransferase MdUGT73AR4, causing it to modify ABA by mass glycosylation and promoting the ABA synthesis pathway, resulting in the accumulation of ABA content, and displaying a stress-resistant phenotype.

## 1. Introduction

Apple (*Malus domestica*) is one of the most widely cultivated fruit trees and is of great economic importance. As the world’s largest producer and consumer of apples, China accounts for more than 40% of the world’s apple cultivation area and production, and it is important to give full play to the advantages of China’s apple industry in order to increase the income of the plantation industry [1]. Since apple production areas are mostly located in arid and semi-arid regions, such as the Loess Plateau, the Bohai Bay, the old course of the Yellow River, and the cool highlands in southwest China, drought has become a major constraint on the growth and development of apples, which seriously affects the sustainable development of the apple industry [2]. Therefore, understanding the drought stress response mechanism of apples is crucial for the apple-growing industry.

Glycosyltransferase (GT) catalyses the glycosylation of a wide range of substances, such as phytohormones, proteins, lipids, and phenylpropanoid compounds, by using activated glycosyl donors as substrates for transfer to small molecules in plants [3]. Studies have shown that glycosyltransferases play an important role in plant cell growth and development, metabolic homeostasis, and other life processes [4]. Based on amino acid sequence similarity, substrate specificity, and catalytic specificity, glycosyltransferases have been classified into 110 different families (CAZy, http://www.cazy.org, accessed on 10 January 2024), of which glycosyltransferase family 1 (GT1) is the most numerous and plays the most important role [5]. The GT1 family primarily catalyses the UDP-glucose to transfer to specific receptors (e.g., proteins, nucleic acids, antibiotics, alkaloids, and phytohormones) and is commonly referred to as UDP-glycosyltransferases (UGT) [4].

Hormones play an important regulatory role in plants, including the regulation of plant responses to various biotic and abiotic stresses. The regulation of hormone levels has wide-ranging effects on cellular metabolic processes, growth and development, and cellular adaptation to environmental changes [6]. Abscisic acid (ABA) is a small-molecule lipophilic phytohormone that is a crucial signalling molecule in plant drought stress [7]. Iuchi et al. found that the *AtNCED3* gene in *Arabidopsis thaliana* was significantly up-regulated in drought stress, and overexpression of the gene into *Arabidopsis thaliana* increased the plant’s ABA content in *Arabidopsis thaliana*, which showed a drought-resistant phenotype [8]. Qian et al. found that exogenous ABA significantly enhanced the drought tolerance of pomegranate by enhancing the metabolic pathways of glycerolipid steroid synthesis, peroxisome biogenesis, photosynthesis, and hemicellulose synthesis by comparing the transcriptomes [9]. Jiang et al. found that exogenous ABA could up-regulate the transcription of related genes and alleviate the inhibitory effect of drought stress on the activities of monodehydroascorbate reductase and dehydroascorbate reductase, which further enhanced the activities of ascorbate peroxidase and glutathione reductase [10].

The regulatory mechanism of glycosylation modification of phytohormone molecules does not require abinitio synthesis, but rather a one-step glycosylation reaction to temporarily activate or block the function of the hormone molecule [11]. Under normal conditions, the content of ABA is in dynamic equilibrium in plants [12]. Therefore, this regulatory mechanism is capable of responding rapidly to signals from inside and outside the cell and returning ABA to endostatic levels quickly after the response. Glycosylation modification of small-molecule hormone analogues is one of the important mechanisms to regulate intracellular hormone levels [13]. It was found that in *Arabidopsis ugt71c5* knockouts, the reduced synthesis of glycosyltransferase UGT71C5 protein resulted in the weakening of ABA glycosylation, resulting in an increase in ABA concentration in the plant, while in overexpressors, the ABA content decreased, indicating that glycosyltransferase UGT71C5 in *Arabidopsis* can regulate abscisic acid concentration through glycosylation modification to maintain its homeostasis in the cell [14].

Under drought stress, plants reduce water dissipation by closing stomata [15]. The plant hormone ABA has an important role in controlling the opening of stomata [16]. Previous studies have shown that the synthesis of ABA increases under drought stress, and the accumulated ABA can induce stomatal closure [17,18,19]. ERECTA family genes regulate stomatal development mainly through the MAPK signalling cascade and affect stomatal development and patterns by phosphorylating to regulate the stability of the transcription factors SPCH, MUTE, and FAMA [20,21]. In this study, we identified MdUGT73AR4, a glycosyltransferase capable of glycosylating ABA, which induces the up-regulation of ABA content through glycosylation modification under drought stress and plays an important role in the closure of the stomata in the lower epidermis of mature leaves, which can lead to the enhancement of drought tolerance and the improvement of yield and quality of dryland apples.

## 2. Results

### 2.1. Identification of the Apple Glycosyltransferase Gene MdUGT73AR4 

Based on the gene chip data published online (https://www.ncbi.nlm.nih.gov/, accessed on 12 August 2023), we found that apple glycosyltransferase genes *MdUGT73AR3* (*Md00G1046200*), *MdUGT73AR4* (*Md07G1007600*), *MdUGT73AR5* (*Md07G1007400*), *MdUGT73AR6* (*XP_008375157.2*), *MdUGT73AB13* (*Md00G1055100*), and *MdUGT73AC7* (*Md05G1085600*) may be involved in drought stress. In order to further identify the target glycosyltransferase genes involved in drought stress, the mRNA expression levels of six glycosyltransferase genes were detected by Real-Time PCR after drought stress, and the results are shown in Figure 1. The apple glycosyltransferase gene *MdUGT73AR4* was up-regulated most significantly (*p* < 0.01). For this reason, we chose the apple glycosyltransferase gene *MdUGT73AR4* as a target gene involved in drought stress.

### 2.2. Analysis of the Expression Pattern of MdUGT73AR4 in Different Parts of ‘Gala’

In order to further understand the tissue expression pattern of *MdUGT73AR4*, we detected the gene expression level of *MdUGT73AR4* in different parts of apples by Real-Time PCR. The results are shown in Figure 2. The mRNA expression level of *MdUGT73AR4* varied in different parts of the apple, with the lowest expression level in stems, and the highest expression level in mature leaves, and it was speculated that *MdUGT73AR4* might function in mature leaves.

### 2.3. HPLC Detection of MdUGT73AR4 Glycosylation Modification of ABA

Glycosyltransferase is a modification class enzyme. In order to verify the glycosylation modification substrate of MdUGT73AR4, we constructed an *MdUGT73AR4* prokaryotic expression vector, transformed it into Escherichia coli strain BL21 for expression, obtained the *MdUGT73AR4* active enzyme protein, and carried out the enzyme reaction in vitro against dozens of small molecule compounds preserved in our laboratory. The results are shown in Figure 3. MdUGT73AR4 can exclusively glycosylate ABA.

### 2.4. Acquisition of MdUGT73AR4 Overexpression Lines and Stomatal Opening Measurement

In the above study, we demonstrated that MdUGT73AR4 can glycosylate and modify ABA in vitro, and then investigated whether MdUGT73AR4 is also able to glycosylate and modify ABA in plants. To solve this problem, we overexpressed *MdUGT73AR4* into apples and obtained transgenic overexpression lines *MdUGT73AR4-OE10* and *MdUGT73AR4-OE15* after screening (Figure 4A). Under drought stress, we measured leaf relative water content and relative conductivity, and the results showed that they significantly increased after drought stress in OE10 and OE15 compared with WT (Figure 4B,C). After exogenous spraying of ABA, we found the ABA content was significantly increased in the overexpression lines, which could be attributed to the accumulation of ABA content due to the glycosylation modification driving the ABA synthesis pathway (Figure 4D). ABA plays an important role in stomatal developmental regulation. The above study also found that *MdUGT73AR4* had the highest level of expression in mature leaves. For this reason, we measured stomatal opening in each line, and the results are shown in Figure 4E, which shows that stomatal opening was decreased in the lower epidermis of mature leaves in the overexpression lines compared with WT. Real-Time PCR was performed to detect the expression of genes related to stomatal opening, and the results show that the regulatory microtubule gene *MdMAP65* was significantly down-regulated in the mature leaves in OE10 and OE15 compared with WT (Figure 4F).

### 2.5. MdUGT73AR4 Upstream Transcription Factors Identified

Bioinformatics analysis of the *MdUGT73AR4* upstream promoter sequence revealed a large number of AREB regulatory elements in the *MdUGT73AR4* promoter sequence (Appendix A). We successfully constructed AREB series elements (AREB1A, AREB1B, AREB2A, AREB2B) into a pGEX-2T vector through a prokaryotic expression vector and transformed them into *E. coli* BL21, purified to obtain active enzyme proteins, and combined them with probes with fluorescence on an *MdUGT73AR4* promoter using an EMSA kit. MdAREB1B binds most tightly to the *MdUGT73AR4* proximal AREB regulatory element (369 bp from the *MdUGT73AR4* start codon ATG) (Figure 5A and Appendix A).

To further verify the binding in vivo, a series of myc-AREB plant overexpression vectors were constructed by linking the AREB series elements to a plant overexpression vector with a myc tag driven by the *MdUGT73AR4* upstream promoter and overexpressed into apple healing tissues by Agrobacterium transformation. Real-Time PCR assay revealed that the *myc-AREB1B* gene was expressed at the highest level (Figure 5B). The target protein AREB1B and the gene *MdUGT73AR4* were found to be in the same complex by ChIP detection, proving that AREB1B is an upstream transcription factor of *MdUGT73AR4*.

## 3. Discussion

Abiotic stress is one of the most susceptible stresses to which plants are subjected, and this stress affects growth and development, signalling, and gene expression [22]. Glycosyltransferases play an important role in the response to stress [23] and have been mainly studied in *Arabidopsis thaliana*. Other species have also been involved but fewer studies have been reported. UDP-glycosyltransferases are known as modulators of small molecule metabolites in plants, which are involved in the response of plants to abiotic stresses by glycosylating small molecules [24]. According to the available reports, there are many glycosyltransferases in *Arabidopsis thaliana* that are involved in plant response to abiotic stresses such as salt, drought, low temperature, high temperature, and low oxygen [14,25,26,27,28,29,30]. Overexpression of *Arabidopsis* IBA glycosyltransferase gene *AtUGT74E2* and cytokinin glycosyltransferase gene *AtUGT76C2* in rice (*Oryza sativa*) can improve salt and drought tolerance [31,32].

The phytohormone ABA plays an extremely important role in plant growth and is a key signal for plant response to abiotic stresses. The study of regulatory modifications in the ABA signalling pathway is increasingly becoming a hot topic. Ubiquitination modifications, epigenetic modifications, and glycosylation modifications of the ABA signalling pathway in plants have been well elucidated [33]. It has been found that OsUGT75A in rice (*Oryza sativa*) promotes submergence tolerance during seed germination [28]. Glycosyltransferase SlUGT75C1 mediates ABA regulation in tomato (*Solanum lycopersicum*) in response to drought stress [34] and *Arabidopsis thaliana* UGT75B1 enhances salt tolerance in seeds and seedlings [35]. However, so far, the involvement of ABA glycosyltransferases in drought stress in apple has been less studied. In this study, we identified for the first time a novel apple glycosyltransferase, MdUGT73AR4, which can modify ABA by glycosylation, affect the ABA content in the plant, regulate stomatal opening, and then participate in the process of drought stress in apple.

It has been shown that the expression of glycosyltransferase genes in various tissues of plants exhibits spatio-temporal specificity [36]. *AtUGT75B1* is expressed in tissues such as germinating seeds, seedlings, root tips, leaves, and flowers, which can be regulated by salt and drought stress, and the level of expression of the gene increases with the prolongation of stress time [37]. *Arabidopsis thaliana AtUGT79B7* negatively regulates hypoxic stress. Real-Time PCR showed that the expression level of the *AtUGT79B7* gene decreased after hypoxic stress treatment [38]. In this study, we also investigated the gene expression characteristics of apple glycosyltransferase MdUGT73AR4 and found that the expression level of *MdUGT73AR4* varied in different parts of apples, with the lowest expression level in the stems and the highest in the mature leaves, so we hypothesised that MdUGT73AR4 might play a role in mature leaves. The “*Gala*” apple samples were treated under drought stress for different times, and the expression level of glycosyltransferase gene *MdUGT73AR4* was detected by Real-Time PCR. It was found that the expression level of the apple glycosyltransferase gene *MdUGT73AR4* showed an increase and then a decrease with the increase in treatment time.

Plant stomata, located on the surface of leaves as well as stems and surrounded by pairs of guard cells, are important sites for CO_2_ uptake, release of photosynthesis-generated O_2_ and H_2_O, and a major channel for water loss through transpiration [39]. When plants sense and respond to environmental stimuli, defence cells can close or open in response to phytohormones and expansion pressures in response to different environmental stimuli, thereby directly controlling the rate and rhythm of CO_2_ uptake and transpirational water loss by regulating the size of the stomatal opening [40,41]. Thus, stomatal movement can influence plant respiration, photosynthesis, and transpiration, which has important implications for plant growth and development, and tolerance to adversity stress. The stress-related hormone ABA is a key signal for plant response to drought stress-induced stomatal closure and plays an important role in the regulation of stomatal development, and *MdUGT73AR4* was also found to have the highest expression level in mature leaves in this study. For this reason, we measured the stomatal opening in each line and found that the stomatal opening in the lower epidermis of mature leaves was decreased in the overexpression lines compared with WT. As an important member of the cytoskeleton, the microtubule skeleton is also involved in the regulation of stomatal movement, and the regulation of stomatal movement is key to the adaptation of plants to environmental stresses. Real-Time PCR was performed to detect the expression of genes related to stomatal opening, and the results are shown that the regulatory microtubule gene *MdMAP65* was significantly down-regulated in the mature leaves in OE10 and OE15 compared with WT. This result suggests that MdUGT73AR4 can regulate stomatal movement through glycosylated ABA, which, in turn, is involved in drought stress.

bZIP is an important class of plant transcription factors that are usually involved in the ABA signalling pathway in response to adversity stresses, such as drought, low temperature, and high salt [42]. bZIP transcription factors recognise cis-acting elements with ACGT as the core sequence, such as elements CACGTG (G-box), CACGTC (C-box), TACGTA (A- box), etc. [43]. These cis-acting elements are involved in ABA-associated stress signalling processes and are ubiquitously present in the promoter regions of stress-responsive genes induced by ABA [44]. In *Arabidopsis thaliana*, overexpression of the bZIP family transcription factors ABF2/AREBI, ABF3, and ABF4/AREB2 all enhanced plant drought tolerance [45]. In addition, overexpression of genes containing ABRE cis-acting elements in *Arabidopsis* (e.g., ANAC002/ATAFl, ANACol9, ANAC055, ANAC072/RD26, GBF3, HISl-3, RD20, RD29B) also enhances plant drought tolerance. In this study, we analysed the *MdUGT73AR4* upstream promoter sequence by bioinformatics and found that the *MdUGT73AR4* promoter sequence has a large number of AREB regulatory elements. To further search for upstream transcription factors, we found that MdAREB1B was most tightly bound to the proximal AREB regulatory element of *MdUGT73AR4* by EMSA experiments. By ChIP assay, we found that the target protein AREB1B and the gene *MdUGT73AR4* are in the same complex, proving that AREB1B is the upstream transcription factor of *MdUGT73AR4*.

In conclusion, this study found that the apple glycosyltransferase MdUGT73AR4 can glycosylate modified ABA and participate in the expression of genes related to regulatory microtubule proteins in mature apple leaves to reduce the stomatal opening of the leaf epidermis and water dissipation to adapt to drought stress. This study is of great significance to enhance the drought resistance of apple and improve the yield and quality of dryland apples.

## 4. Materials and Methods

### 4.1. Plant Material

The plant material used in this study was “*Gala*” apple in vitro seedlings, which were bred and preserved in our laboratory. The seedlings were grown under a photoperiod of 16 h/8 h (light/dark), a light intensity of 40 μmol·m^−2^·s^−1^, and a day/night temperature of 25 °C/20 °C. Fresh materials from different parts of apple were collected from roots, stems, leaves, flowers, and fruits of 10-year-old “*Gala*” apple (*Malus × domestica*) trees planted in the College of Life Sciences, Shandong Agricultural University. After sampling, the samples were snap-frozen in liquid nitrogen and stored at −80 °C.

### 4.2. Apple Seedling Stress Treatment

Apple seedlings, which had been cultured normally for 2 months, were randomly divided into 2 groups: the control group continued to be cultured normally and was watered with 1/2 MS nutrient solution; the drought stress group was watered with a nutrient solution containing 200 mmol/L mannitol and 1/2 MS nutrient solution for 3, 6, 12, and 24 h. At the end of the treatments, the samples were quickly frozen with liquid nitrogen and stored in a refrigerator at −80 °C.

### 4.3. Real-Time PCR

Total RNA was extracted according to the procedure of the Plant RNA Kit (R6827-00, Omega Bio-Tek, Norcross, GA, USA), and cDNA was synthesised using TransScript^®^II First-Strand cDNA Synthesis SuperMix (AT301-02, Alltech Gold, Beijing, China). Real-Time PCR was performed using an SYBR Green PCR Master Mix kit (Q111-02, VazymE, Nanjing, China) by a Bio-Rad Thermal Cycling System (CFX Connect, Hercules, CA, USA). The apple *MdUBQ* gene was used as the internal reference gene, and primers for other genes were designed with Primer premier 5.0 software, and synthesised by Sangon Bioengineering Co. (Shanghai, China) (Appendix A).

The 10 µL reaction system consisted of 0.4 µL of each of the upstream and downstream primers, 5 µL of MonAmp™ ChemoHS qPCR Mix (MQ00401, Monad Biotechnology Co., Ltd., Suzhou, China), 0.1 µL of Low ROX Dye, 1–2 µL of cDNA template, and nuclease-free water to bring the volume to 10 µL.

The PCR amplification procedure consisted of 95 °C for 5 min, 40 cycles, denaturation at 95 °C for 5 s, annealing at 60 °C, and extension for 34 s. The relative expression of the genes was calculated using the 2^−ΔΔCt^ formula. Each assay was biologically replicated at least 3 times and the average value was taken.

### 4.4. Vector Construction

The cDNA of apple seedlings was used as the template for PCR amplification (95 °C for 5 min; 95 °C for 45 s; 55 °C for 30 s; 72 °C for 45 s, 35 cycles; 72 °C for 4 min), and the full-length target band of *MdUGT73AR4* was obtained (Appendix A). The target fragment was recovered by a DNA gel recovery kit (ZY511-100, Runye Biotechnology Co., Shanghai, China) and then ligated into a pMD19-T cloning vector (vector:target DNA = 1:4) at 16 °C for 30 min, and then sent to Shanghai Sangon Biotechnology Co., Ltd. for sequencing after transforming and double-enzymatic digestion of the positive bacteria. One hundred percent correct sequencing was performed, and then the *MdUGT73AR4* gene fragment was recombinantly inserted into prokaryotic expression vector pGEX-2T (Appendix A) and plant expression vector PBI121 (Appendix A), respectively. Then, the positive recombinant plasmids MdUGT73AR4-pGEX-2T and MdUGT73AR4-PBI121 were transformed into *E. coli* BL-21 and R. rhizogenes strain K599 for subsequent protein expression and transgenesis, as verified by PCR and double enzyme digestion, respectively, which were used for subsequent protein expression and transgenic strain acquisition.

### 4.5. MdUGT73AR4 Glycosylation Reaction

Protein expression: the MdUGT73AR4-pGEX-2T BL-21 strain containing the positive plasmid was activated, amplified, and when its OD value was about 0.6, 0.5 mol/L IPTG was added to make a final concentration of 0.5 mmol. After induction, the target protein was purified by using the GST-tagged Protein Purification kit (P2262, Biyuntian Biotechnology Co., Shanghai, China) and used for the enzyme activity assay. The amino acid sequence is shown in Appendix A.

Enzymatic reaction in vitro: 10 µL of 0.5 M Tris-HCl (pH 8.0), 5 µL of 50 mM MgSO_4_, 5 µL of 200 mM KCl, 2.5 µL of 0.1 M UDP-glucose, 1 µL of 10% β-mercaptoethanol, 1 µL of 100 mM ABA, 2 µL of MdUGT73AR4 protein, and ddH_2_O were added to the enzymatic system to make the reaction volume 100 µL.

### 4.6. Acquisition of Transgenic Plants

The apple genetic transformation experiment used “*Gala*” apple in vitro seedlings as materials and obtained “GL-3” seedlings with strong regeneration ability and high transformation efficiency. The leaves of “GL-3” were transformed with Agrobacterium tumefaciens carrying the sequence of *MdUGT73AR4*, and the transgenic apple lines OE10 and OE15 were obtained.

DNA level identification of transgenic lines: DNA was extracted from “GL-3” and transgenic apple plants using the Plant Genomic DNA Extraction Kit (D9194, TaKaRa, Bao Bioengineering Co., Ltd., Dalian, China), and part of the sequence of the CaMV 35S promoter of the vector was used as the upstream primer, and the sequence of the *MdUGT73AR4* gene was used as the downstream primer for PCR amplification. The primers are shown in Appendix A. Real-Time PCR was used to identify *MdUGT73AR4*-positive lines.

Identification of mRNA levels in transgenic lines: RNA was extracted from *MdUGT73AR4* transgenic and non-transgenic plants, respectively, using the SPARKeasy Polysaccharide Polyphenol/Complex Plant RNA Rapid Extraction Kit (AC0307, SparkJade Biotechnology Co., Ltd., Shandong, China) to reverse transcription of RNA to cDNA. Real-Time PCR was used to detect the expression of *MdUGT73AR4* in positive plants with the primers shown in Appendix A. The identified *mdugt73ar4*-positive transgenic plants and untransformed wild-type plants (“GL-3”) were propagated for subsequent experimental treatments.

### 4.7. Extraction and HPLC Analysis of ABA Glycosides in Apple Seedlings

Exogenous ABA treatment: WT “*Gala*” and transgenic apple lines (OE10 and OE15), which had been cultured normally for 2 months, were randomly divided into 2 groups, the control group, which continued to be cultured normally with 1/2 MS nutrient solution, and the treatment group, which was treated with 10 µmol/L ABA for 12 h.

Extraction of ABA glycosides from apple seedlings: 10 g of fresh sample was taken and ground into powder with liquid nitrogen, and 10 mL of 80% methanol solution was added for extraction, along with 2 µL of internal ginseng Picloram (1918-02-1, Xingkaiyue Biotechnology Co., Shenzhen, China). The sample was shaken well and allowed to stand at room temperature for 6 h. During this time, the sample was mixed by continuous inversion. After vacuum evaporation in a rotary evaporator, the residual powder on the inner wall of the centrifuge tube was dissolved with 1 mL of methanol solution and centrifuged at 15,000× *g* rpm for 20 min, and the glycosides were detected by HPLC.

HPLC analysis: A Shimadzu LC-20AT analyser (Shimadzu, Japan) was used. The main instrumentation consisted of a workstation LC solution (Verl.21), a diode array detector SPD-M20A, an autosampler SIL-20A, a system controller CBM-20A, and a degasser DGM-20A3. The column was a reversed-phase column. The mobile phases were acetonitrile and water (both containing 0.1% trifluoroacetic acid) at a flow rate of 1 mL/min with an elution time of 35 min and a detection wavelength of 320 nm. The detection wavelengths of the various elution peaks were between 190 and 430 nm.

LC-MS (Liquid Chromatography–Mass Spectrometry) analysis: a USA Surveyor MSQ single quadrupole column Liquid Chromatography–Mass Spectrometer (BRE725539, Thermo Fisher Scientific, Shanghai, China) was used as the instrument, and an ESI (electrospray ionisation) interface was used in this experiment. The mobile phase was 0.1% formic acid instead of 0.1% phosphoric acid or triethylaminoacetic acid to reduce ion suppression. The chromatographic conditions were the same as the HPLC elution method, and the detection was carried out in positive ion mode with a surface-induced decomposition intensity of 30 eV.

### 4.8. Measurement of Stomatal Opening

Each apple strain was taken after exogenous ABA treatment, and the fully expanded leaves were picked and quickly washed with distilled water. The lower epidermis was torn with tweezers, and after removing the chloroplasts with a bristle brush, the leaves were placed in Mes-KCl buffer (10 mmol/L Mes, 50 mmol/L KCl, 0.1 mmol/L CaCl_2_, pH 6.24) and the stomatal openings were observed under an inverted microscope (10 × 40). The size of the stomatal openings was observed. At least three epidermal strips were observed for each treatment, four fields of view were randomly selected for each epidermal strip, ten stomatal openings were randomly measured for each field of view, and each treatment was repeated at least eight times (not fewer than four hundred stomata). The mean and standard error were calculated.

### 4.9. Measurement of Relative Water Content and Relative Conductivity of Leaves as Indicators of Drought Tolerance Physiology

When all the test apple materials were 70–80 cm in height, plants of uniform height were selected, fully matured leaves in the middle of each plant were collected, and the material was taken at different time points (0, 20, 40, 60, 80, 100, 120, 180, 240, 300, 360, 420, 480 min) in the isolated state for the determination of the relative water content and relative conductivity.

Determination of leaf relative water content: After weighing the fresh mass of the leaf blade (FW), the leaf blade was immersed in distilled water for 24 h. After absorbing the surface water with absorbent paper, the saturated mass of the leaf blade (TW) was measured, the dry mass of the leaf blade (DW) was measured after drying to a constant weight, and the leaf relative water content was calculated, with five replications for each germplasm resource. The formula for calculating leaf relative water content was as follows: relative water content (RWC)/% = (FW − DW)/(TW − DW) × 100%.

Measurement of relative conductivity: 20 discs were punched on the leaves with a perforator, avoiding the veins, and then placed into a 15 mL centrifuge tube with 10 mL of purified water and soaked for 4 h. The conductivity was measured by a conductivity meter (S1), then in a boiling water bath for 20 min, then cooled down to room temperature and mixed again to measure the conductivity (S2), and then in purified water (S0). To calculate the relative conductivity of the leaves, five replicates were made in each germplasm. The relative conductivity was calculated as follows: REC/% = (S1 − S0)/(S2 − S0) × 100%.

### 4.10. EMSA and ChIP Experiments

EMSA experiments: 3′-biotin-labelled probes and unlabelled probes (cold probes) were synthesised by Sangon Bioengineering Co. (Shanghai, China), and the corresponding probe sequences are shown in Appendix A. The synthesised probes were first annealed and complemented into double-stranded probes for subsequent EMSA experiments. Subsequently, the binding of MdUGT73AR4 protein to the probe was verified using the LightShiftTM EMSA optimisation and control kit (20148X, Thermo scientific, Waltham, MA, USA) with reference to the instructions for use.

ChIP experiments: A ChIP Assay Kit was used (P2078, Shanghai Biyuntian Biotechnology Co., Ltd., Shanghai, China) according to the instructions of the kit and stored at 4 °C for spare use.

### 4.11. Statistical Analyses

SPSS Statistics for Windows version 17.0 (SPSS Inc., Chicago, IL, USA) was used for statistical analyses. Statistical significance was assessed using one-way ANOVA and Tukey’s multiple-range test. Different letters indicate significant differences (*p* < 0.05) and highly significant differences (*p* < 0.01).

## Figures and Tables

**Figure 1 ijms-25-05672-f001:**
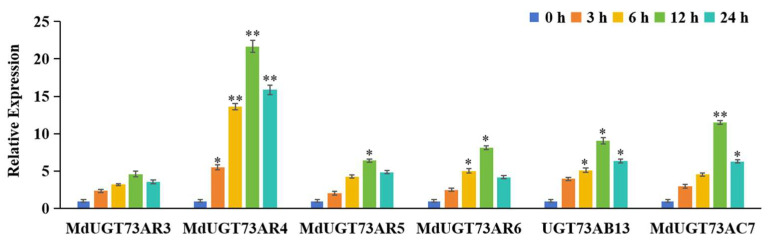
Real-Time PCR detection of glycosyltransferase genes *MdUGT73AR3* (*Md00G1046200*), *MdUGT73AR4* (*Md07G1007600*), *MdUGT73AR5* (*Md07G1007400*), *MdUGT73AR6* (*XP_008375157.2*), *MdUGT73AB13* (*Md00G1055100*), and *MdUGT73AC7* (*Md05G1085600*) induced expression of mRNA levels in “Gala” apple sample seedlings after drought stress treatment for 0 h, 3 h, 6 h, 12 h, and 24 h. Note: * *p* < 0.05, ** *p* < 0.01, *n* = 3.

**Figure 2 ijms-25-05672-f002:**
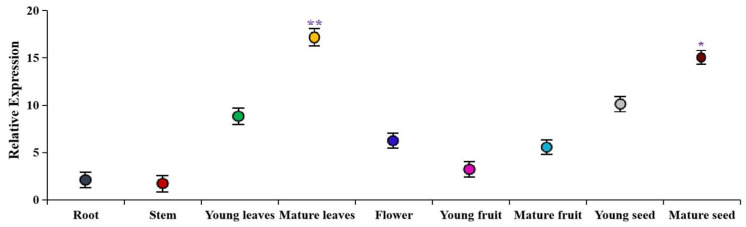
Expression pattern of different parts of apple glycosyltransferase gene *MdUGT73AR4* detected by Real-Time PCR. Note: * *p* < 0.05, ** *p* < 0.01, *n* = 3.

**Figure 3 ijms-25-05672-f003:**
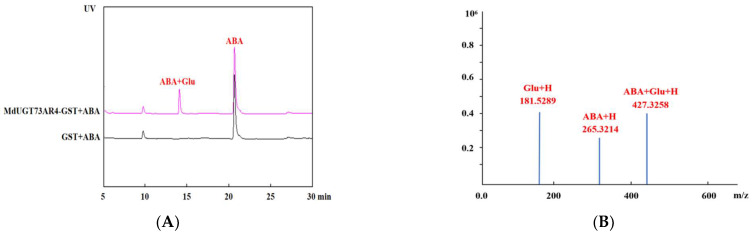
HPLC detection of MdUGT73AR4 glycosylation modification substrate ABA. (**A**): HPLC detection of MdUGT73AR4 glycosylation modification product ABA glucoside; (**B**): mass spectrometry detection of MdUGT73AR4 glycosylation modification product ABA glucoside molecular weight.

**Figure 4 ijms-25-05672-f004:**
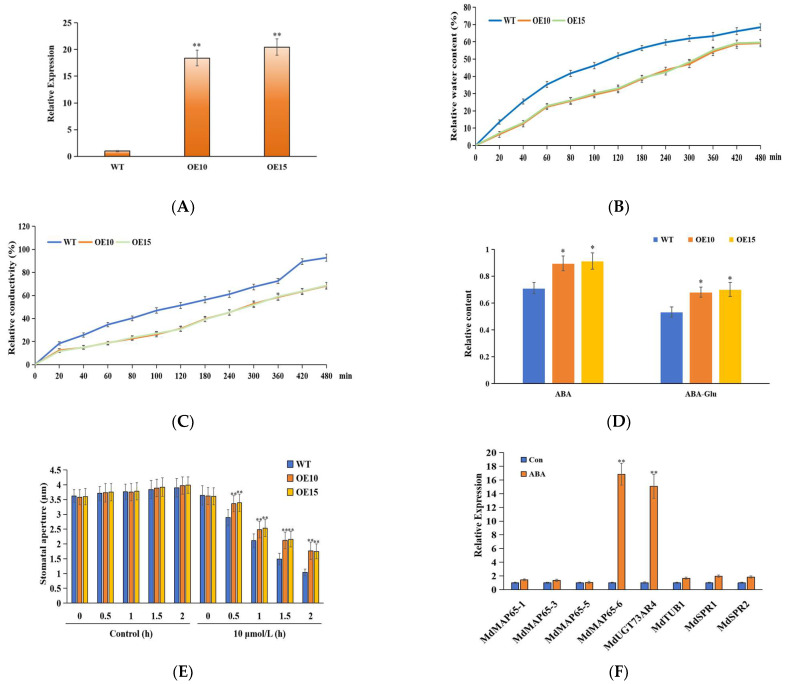
Obtained *MdUGT73AR4* overexpression lines and phenotypic observation. (**A**) Detection of gene *MdUGT73AR4* mRNA expression level in overexpression lines; (**B**) determination of relative water content in mature leaves of *MdUGT73AR4* overexpression lines OE10 and OE15; (**C**) determination of relative conductivity in mature leaves of *MdUGT73AR4* overexpression lines OE10 and OE15; (**D**) detection of ABA and ABA glycoside contents in *MdUGT73AR4* overexpression lines OE10 and OE15; (**E**) determination of stomatal opening in *MdUGT73AR4* overexpression lines OE10 and OE15 after drought stress; (**F**) determination of stomatal expression levels of stomatal-related genes in *MdUGT73AR4* overexpression lines OE10 and OE15 after drought stress. Note: * *p* < 0.05, ** *p* < 0.01, *n* = 3.

**Figure 5 ijms-25-05672-f005:**
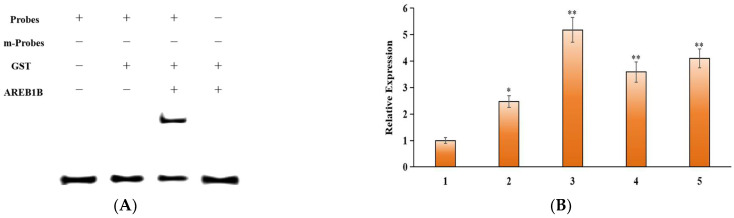
*MdUGT73AR4* upstream transcription factor identification. (**A**) EMSA identification of *MdUGT73AR4* upstream transcription factor AREB1B; (**B**) ChIP assay identification of *MdUGT73AR4* upstream transcription factor AREB1B. 1, MYC; 2, AREB1A; 3, AREB1B; 4, AREB2A; 5, AREB2B. Note: * *p* < 0.05, ** *p* < 0.01, *n* = 3.

## Data Availability

Data will be made available upon request.

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
