# Peer review of "Apple Glycosyltransferase MdUGT73AR4 Glycosylates ABA to Regulate Stomatal Movement Involved in Drought Stress"

_ijms, 2024, doi:10.3390/ijms25115672_

Round 1
Reviewer 1 Report
Comments and Suggestions for Authors
The manuscript " ‘Apple glycosyltransferase MdUGT73AR4 glycosylates ABA to regulate stomatal movement involved in drought stress" by Mu et al. (ijms-301330) aims at analysis function of MdUGT73R4 response to drought stree in apple, it is useful in apple industry. However, there are several issues that prevent me from recommending the publication of the manuscript in the current version.
First of all, Six genes showed similar expression trends in Figure 1., MdUGT73AR4 was chosed to be candidate gene due to it has higher expression, is this decision scientific ?
Second, Some Figures are not clear enough, for example, Figure 4E and Figure 4F.
Third, It is lack of drought tolerant phenotyp for transgenic lines, could you add it ?
A few other minor revions suggested:
-line 13: ‘Real time PCR’ should be ‘Real-Time PCR’.
-line 14: ‘in the mature leaves’ should be ‘in drought stress’.
-line 84-90: These genes have little correlation with the content of the research presented below, and the coherence needs to be improved.
-line 108-112: What do 0 h, 3 h, 6 h, 12 h and 24 h stand for? Is it drought treatment time? Need to indicate in the note of Figure 1.
-line 113: ‘Analysis of the expression pattern of different parts of MdUGT73AR4’ should be revised to ‘Analysis of the expression pattern of MdUGT73AR4 in different parts of ‘Gala’’.
-line 119: High expression of MdUGT73AR4 have function, low expression don’t have function ?
-line 137: ‘MdUGT73AR4’ should be ‘MdUGT73AR4’, the name of Protein can’t be italic, please check others in the manuscript.
-line 167-175: What do 0, 20, 40, 60, 80, 100, 120 ,180…. stand for? Need to indicate in the note of Figure 4B, C.
-line 206: fewer studies have been reported in apples ?
-line 209: removed ‘in plant’.

Author Response
Dear Editors and Reviewers:
Thank you for your letter and for the reviewers’ comments concerning our manuscript entitled “Apple glycosyltransferase MdUGT73AR4 glycosylates ABA to regulate stomatal movement involved in drought stress”(Manuscript Number: ijms-301330). Those comments are all valuable and very helpful for revising and improving our paper, as well as the important guiding significance to our researches. We have studied comments carefully and have made correction which we hope meet with approval. Revised portion are marked in red in the paper. The main corrections in the paper and the responds to the editor’s and reviewer’s comments are as following:
Responds to the reviewer’s comments:
Response to Reviewer 1 Comments
Point 1: Six genes showed similar expression trends in Figure 1., MdUGT73AR4 was chosed to be candidate gene due to it has higher expression, is this decision scientific ?
Response 1: Thank you very much for your valuable suggestions. We chose the six genes in the Figure 1, because we previously found that all six genes belong to the Group D family of GT1 (Li et al. 2022), and the bioinformatics functional analysis predicted that all six genes function may be involved in drought stress, but these genes have not yet been validated, after our apple seedlings with drought stress, the expression of these genes were determined by Real-time PCR, found that the induced up-regulation of MdUGT73AR4 was most significant under drought stress conditions. Therefore, we suspect that this gene plays the most obvious role under drought stress, which can contribute to the subsequent breeding studies of hartolerant apples, so choosing MdUGT73AR4 as a candidate gene. Thank you again!
Point 2: Some Figures are not clear enough, for example, Figure 4E and Figure 4F.
Response 2: Thank you very much for your valuable suggestions. We have adjusted the Figure 4E and Figure 4F. Thank you again!
Point 3: It is lack of drought tolerant phenotyp for transgenic lines, could you add it ?
Response 3: Thank you very much for your valuable suggestions. Analysis of spatiotemporal expression patterns revealed that the glycosyltransferase gene MdUGT73AR4 acted predominantly in mature leaves. In this study, MdUGT73AR4-OE10 and OE15 were overexpressed at a level of about 20-fold, and the overexpression lines did not show a very significant drought stress phenotype, which may be related to the expression site and expression level of this gene. Thank you again!
Point 4: -line 13:‘Realtime PCR’should be‘Real-Time PCR’.
Response 4: Thank you very much for your valuable suggestions. It’s our negligence, we have changed‘Realtime PCR’to‘Real-Time PCR’and marked it in red. Thank you again!
Point 5: -line 14: ‘in the mature leaves ’ should be ‘in drought stress ’.
Response 5: Thank you very much for your valuable suggestions. We have replaced‘in the mature leaves’with‘in drought stress’and marked it in red. Thank you again!
Point 6: -line 84-90: These genes have little correlation with the content of the research presented below, and the coherence needs to be improved.
Response 6: Thank you very much for your valuable suggestions. To ensure contextual relevance and coherence of the research context, we have removed line 84-90‘It was found that Arabidopsis ABA synthesis mutants and signalling-deficient mutants displayed higher stomatal densities compared with the wild type (WT), wherea fewer stomata were found in ABA catabolism-deficient mutants [21], suggesting that ABA is able to inhibit stomatal development. In cereals, drought tolerance was positively regulated by the SiYTH1 gene, which was revealed by Luo et al. The siyth1 mutant exhibited reduced stomatal closure under drought stress [22]. Previous studies have demonstrated that ERECTA family genes have key functions in regulating stomatal development in plants [23–25].’and marked it in red. The serial number of the reference 21-25 has been deleted and the serial number after the original document 25 has been changed accordingly, which has been marked in red in the manuscript. Thank you again!
Point 7: -line 108- 112: What do 0 h, 3 h, 6 h, 12 h and 24 h stand for? Is it drought treatment time? Need to indicate in the note of Figure 1.
Response 7: Thank you very much for your valuable suggestions. 0 h, 3 h, 6 h, 12 h and 24 represent the drought treatment time, which we have annotated in red in the note of Figure 1. Thank you again!
Point 8: -line 113: ‘Analysis of the expression pattern of different parts of MdUGT73AR4 ’ should be revised to ‘Analysis of the expression pattern of MdUGT73AR4 in different parts of ‘Gala ’’ .
Response 8: Thank you very much for your valuable suggestions. We have replaced‘Analysis of the expression pattern of different parts of MdUGT73AR4’with‘Analysis of the expression pattern of MdUGT73AR4 in different parts of‘Gala’’and marked it in red. Thank you again!
Point 9: -line 119: High expression of MdUGT73AR4 have function, low expression don’t have function ?
Response 9: Thank you very much for your valuable suggestions. The low gene expression sites were functional, but the high gene expression sites had a more pronounced drought tolerance phenotype. Thank you again!
Point 10: -line 137: ‘MdUGT73AR4 ’ should be ‘MdUGT73AR4’, the name of Protein can’t be italic, please check others in the manuscript.
Response 10: Thank you very much for your valuable suggestions. We have changed 'MdUGT73AR4' to 'MdUGT73AR4' in the 137 line and marked it in red, we have checked the other content of the article, and changed the 'MdUGT73AR4' of article line 381 and line 470 to 'MdUGT73AR4' and marked it in red. Thank you again!
Point 11: -line 167- 175: What do 0, 20, 40, 60, 80, 100, 120 ,180 … . stand for? Need to indicate in the note of Figure 4B, C.
Response 11: Thank you very much for your valuable suggestions. We are very sorry, it’s our negligence, and the abscissa 0,20,40,60,80,100,120,180… represent the time in isolation after the leaf in minutes. Thank you again!
Point 12: -line 206: fewer studies have been reported in apples ?
Response 12: Thank you very much for your valuable suggestions. Glycosyltransferase genes have been poorly studied in apples, because of the relative growth cycle of plants longer. Thank you again!
Point 13: -line 209: removed ‘in plant ’.
Response 13: Thank you very much for your valuable suggestions. We have removed the 'in plant' at line 209, and marked it in red. Thank you again!
Special thanks to you for your good comments.
We tried our best to improve the manuscript and made some changes in the manuscript. These changes will not influence the content and framework of the paper. And here we did not list the changes but marked in red in revised paper.
We appreciate for Editors/Reviewers’ warm work earnestly, and hope that the correction will meet with approval.
Once again, thank you very much for your comments and suggestions.
Yours sincerely,
Pan Li
Institution and address: Liaocheng University
E-mail: lipan202301@163.com

Reviewer 2 Report
Comments and Suggestions for Authors
The manuscript contains interesting research results with abscisic acid (ABA) and its involvement in plant response to drought stress. I ask the authors to indicate the possibility of using the obtained research results in practice. I also ask for indications of future research directions.
The manuscript is very carefully prepared. Congratulations to the Authors!
Comments
Line 291, in what year was the research done? In what research center?
Line 291, please provide the name of the breeder of the studied variety and its brief characteristics
References, I suggest removing publication 23, which is more than 10 years old.
Author Response
Dear Editors and Reviewers:
Thank you for your letter and for the reviewers’ comments concerning our manuscript entitled “Apple glycosyltransferase MdUGT73AR4 glycosylates ABA to regulate stomatal movement involved in drought stress”(Manuscript Number: ijms-301330). Those comments are all valuable and very helpful for revising and improving our paper, as well as the important guiding significance to our researches. We have studied comments carefully and have made correction which we hope meet with approval. Revised portion are marked in red in the paper. The main corrections in the paper and the responds to the editor’s and reviewer’s comments are as following:
Responds to the reviewer’s comments:
Response to Reviewer 2 Comments
Point 1: The manuscript contains interesting research results with abscisic acid (ABA) and its involvement in plant response to drought stress. I ask the authors to indicate the possibility of using the obtained research results in practice. I also ask for indications of future research directions.The manuscript is very carefully prepared. Congratulations to the Authors!
Response 1: Thank you so much for your approval, and for your valuable suggestions. In this study, apple drought tolerance can be improved by overexpressing MdUGT73AR4, which will provide a new research direction for cultivating drought-tolerant apples and help to improve the yield and quality of apples. Thank you again!
Point 2: Line 291, in what year was the research done? In what research center?
Response 2: Thank you very much for your valuable suggestions. The study was completed in the Laboratory of Molecular Biology, School of Pharmacy, Liaocheng University in 2023. Thank you again!
Point 3: Line 291, please provide the name of the breeder of the studied variety and its brief characteristics
Response 3: Thank you very much for your valuable suggestions. The apple variety used in this study was Malus × domestica, the 10-year-old 'Gala' apple tree was by the College of Life Sciences of Shandong Agricultural University, and the apple seedlings used in the experiment were cultivated by Lijun Mu. Gala apples are native to New Zealand and were bred in 1939 by New Zealand fruit breeder Kidd. The main characteristics of this species are short, thick, cylindrical, densely hairy when young, and purple-brown and glabrous on the old branches; Winter buds ovate, apex obtuse, densely pubescent. The leaves are elliptic, ovate to broadly elliptic, the fruit is oblate and spherical, the apex is often raised, the sepals are sunken, the sepals are permanent, and the peduncle is short and thick. Thank you again!
Point 4: References, I suggest removing publication 23, which is more than 10 years old.
Response 4: Thank you very much for your valuable suggestions. We have removed publication 23 and marked it in red. Thank you again!
Special thanks to you for your good comments.
We tried our best to improve the manuscript and made some changes in the manuscript. These changes will not influence the content and framework of the paper. And here we did not list the changes but marked in red in revised paper.
We appreciate for Editors/Reviewers’ warm work earnestly, and hope that the correction will meet with approval.
Once again, thank you very much for your comments and suggestions.
Yours sincerely,
Pan Li
Institution and address: Liaocheng University
E-mail: lipan202301@163.com
